# Medical, Endoscopic and Surgical Management of Stricturing Crohn’s Disease: Current Clinical Practice

**DOI:** 10.3390/jcm11092366

**Published:** 2022-04-23

**Authors:** Fotios S. Fousekis, Ioannis V. Mitselos, Kostas Tepelenis, George Pappas-Gogos, Konstantinos H. Katsanos, Georgios D. Lianos, Francesco Frattini, Konstantinos Vlachos, Dimitrios K. Christodoulou

**Affiliations:** 1Department of Gastroenterology and Hepatology, University Hospital of Ioannina, University of Ioannina, 45110 Ioannina, Greece; fotisfous@gmail.com (F.S.F.); jmitselos@gmail.com (I.V.M.); khkostas@hotmail.com (K.H.K.); 2Department of Surgery, University Hospital of Ioannina, University of Ioannina, 45110 Ioannina, Greece; kostastepelenis@gmail.com (K.T.); pappasg8@gmail.com (G.P.-G.); georgiolianos@yahoo.gr (G.D.L.); kvlachos@uoi.gr (K.V.); 3Department of Surgery, ASST Settelaghi, 21100 Varese, Italy; francescofrattini79@gmail.com

**Keywords:** Crohn’s disease, strictures, stenoses, anti-TNF, endoscopic balloon dilatation, surgery

## Abstract

The development of fibrostenotic intestinal disease occurs in approximately one-third of patients with Crohn’s disease and is associated with increased morbidity. Despite introducing new biologic agents, stricturing Crohn’s disease remains a significant clinical challenge. Medical treatment is considered the first-line treatment for inflammatory strictures, and anti-TNF agents appear to provide the most considerable benefit among the available medical treatments. However, medical therapy is ineffective on strictures with a mainly fibrotic component, and a high proportion of patients under anti-TNF will require surgery. In fibrotic strictures or cases refractory to medical treatment, an endoscopic or surgical approach should be considered depending on the location, length, and severity of the stricture. Both endoscopic balloon dilatation and endoscopic stricturoplasty are minimally invasive and safe, associated with a small risk of complications. On the other hand, the surgical approach is indicated in patients not suitable for endoscopic therapy. This review aimed to present and analyze the currently available medical, endoscopic, and surgical management of stricturing Crohn’s disease.

## 1. Introduction

Crohn’s disease (CD) is a systemic disease, primarily affecting the gastro-intestinal tract, of which the etiology has not been clarified. Genetic and environmental factors are indicated to contribute to CD development. CD may affect any part of the digestive tract, whereas disease behavior may change over time and progress to fibrostenotic and penetrating disease. It is estimated that approximately 40% of CD patients will develop naïve symptomatic strictures (e.g., intestinal obstruction), whereas it is not uncommon for the development of anastomotic strictures. Diagnosis <40 years of age, need for steroids at diagnosis, small bowel involvement, and smoking has been associated with stricturing CD. Symptoms of stricturing CD may include nausea, vomiting, abdominal cramping, and abdominal flatulence, leading to decreased quality of life [1].

An expert consensus defined small bowel stricture on imaging (Computed Tomography Enterography, Magnetic Resonance Enterography, and ultrasound) as “a localized luminal narrowing and bowel wall thickening with pre-stricture dilation”. Pre-stricture (or prestenotic) dilation is defined as a greater-than 3 cm bowel lumen diameter before the stenotic segment. Endoscopically, a naïve stricture is defined as the “Inability to pass an adult endoscope through the narrowed area without prior endoscopic dilation and with a reasonable amount of pressure applied”. On the other hand, successful treatment of a small bowel stricture is characterized by relief of clinical symptoms and improvement of endoscopic/radiologic features [2].

CD-induced intestinal strictures may be differentiated into inflammatory, fibromatous, and mixed types. Patients with strictures with a predominately fibrotic component were not shown to benefit from medical management, which is only effective in patients with inflammatory intestinal strictures. In this case, endoscopic or surgical approaches may provide an efficient alternative therapeutic solution [3]. However, differentiating fibrotic from inflammatory strictures is challenging, as most imaging techniques, such as CT or MR enterography, show limitations regarding the discrimination of fibrotic and inflammatory strictures. Nevertheless, the combination of novel imaging techniques, such as contrast-enhanced ultrasound and ultrasound elasticity imaging, appear to be promising diagnostic tools [4].

On the other hand, inflammatory markers, such as CRP and fecal calprotectin, may reflect CD activity, contributing to distinguishing between fibrotic and inflammatory strictures without being strictly specific [5]. Thus, clinicians should take into account the clinical course of CD, inflammatory biomarkers, and imaging techniques in order to diagnose a fibrotic or inflammatory stenosis.

The ongoing development of new biological agents offers more therapeutic options with different action mechanisms, however; the management of stricturing CD remains a challenging task. The purpose of this narrative review is to present, analyze and compare the current medical, endoscopic, and surgical approaches of stricturing CD treatment.

## 2. Literature Research

We performed an in-depth review of the literature in PubMed to identify articles about the management of stricturing CD, using the following search string (“stricturing Crohn’s disease” OR “stenotic Crohn’s disease” OR “fibrostenotic Crohn’s disease”) AND (“medical management” OR “anti-TNF” OR “vedolizumab” OR “ustekinumab” OR “endoscopic management” OR “surgical management”). Furthermore, the references of relevant papers were also reviewed. Only articles in the English language were used. We extensively examined the abstracts of manuscripts and identified the most relevant articles.

## 3. Medical Management

CD patients with suspicion of intestinal obstruction should be hospitalized and be treated with bowel rest, intravenous fluid, and electrolyte replacement. Traditionally, intravenous corticosteroids may relieve symptoms of intestinal obstruction, reducing transmural edema and increasing the luminal area in patients. Corticosteroids should be administered in patients without signs of peritonitis and suspicious of perforation. A surgical approach is required if conservative management is failed and/or there are features of abscess, phlegmon, and malignancy. In addition, early bowel resection could be an option in high-risk patients with isolated ileocecal stricturing CD, as this practice is demonstrated to offer better outcomes when compared to medical treatment regarding the subsequent development of fistula or intestinal obstruction [6].

As mentioned above, medical treatment is effective on inflamed strictures, whereas anti-fibrotic intestinal therapy is lacking. Several studies have been suggested the beneficial effect of anti-TNF agents on established stricturing CD (Table 1). In a prospective study with CD patients suffering from small intestinal strictures, treatment with infliximab ceased the progression of strictures in 3 out of 15 patients, while 6 patients (demonstrated regression of stenoses [7]. A randomized control trial demonstrated no increased risk of strictures development in CD patients receiving infliximab [8]. Furthermore, multivariable analysis of a cohort study found that infliximab use was not associated with a higher risk of intestinal stricture or obstruction [9]. A recent multicenter retrospective study demonstrated that one-quarter of patients with stricturing CD benefited from the use of infliximab or adalimumab [10]. Another multicenter study indicated that more than half of the patients with stricturing CD were free of surgery after initiation of adalimumab administration. At week 24, 64% of patients were steroid-free [11]. An observational retrospective study documented that approximately two-thirds of patients with colonic or small bowel stenotic CD avoided the surgery by receiving adalimumab or infliximab, whilst risk factors for abdominal surgery were the existence of non-perianal fistula (HR: 9.77, 95% CI: 2.99–31.9, *p* < 0.001) and pre-stenotic dilation (per 1 mm increase, HR: 1.08, 95% CI: 1.01–1.15, *p* = 0.022) [12].

Another study analyzing the efficacy of anti-TNF agents (adalimumab and infliximab) on patients with stricturing CD showed that 66.6% of patients required an endoscopic or surgical approach, hospitalization, or treatment discontinuation within 60 months after anti-TNF administration. CD diagnosis after 40 years old, small stricture luminal diameter, increased stricture wall thickness, and fistula with abscess were negative predictor factors, while anti-TNF combination therapy with immunomodulators was associated with a better prognosis [13]. Thus, there is increased heterogenity among studies about the role of anti-TNF agents on stricturing CD, because the length of follow-up intervals, inclusion criteria, characteristics of patients and the end-points vary. However, it seems that a portion of patients with stricturing CD seems to respond to anti-TNF therapy, while infliximab administration appears to be not associated with strictures development.

Ustekinumab is a monoclonal IgG1 antibody against interleukin-12 (IL-12) and IL-23 that may be used as second and third-line treatment in CD patients, while some authors recommend ustekinumab as a first-line treatment in patients with CD and psoriasis [14]. However, data regarding the effect of ustekinumab on stricturing CD is limited. Analysis from a nationwide cohort study found that stricturing CD appears to be no predictor factor for corticosteroid-free remission at week 52 in CD patients receiving ustekinumab [15]. On the other hand, there is also limited data about the efficacy of vedolizumab in stricturing CD. In a multicenter cohort study, 118 out of 224 (55.7%) patients suffered from stricturing or penetrating CD, and after 12 months, 3 patients underwent subtotal colectomy for colonic stricture, and 2 patients needed surgical resection for small bowel strictures [16]. 

**Table 1 jcm-11-02366-t001:** Studies about medical therapy of stricturing Crohn’s disease.

Study Design	Agent	Number of CD Patients	Follow-Up Period	Results	Study (Year of Study)
Prospective	Infliximab	15 Pts with small bowel stenoses received IFX	Median 38 months, range 7–58 months	5 Pts Stop treatment	Pallota N et al. [7] (2008)
				3 Pts: No progression of stenoses	
				6 Pts: Regression of stenosis	
RCT (ACCENT I)	Infliximab	188 Pts received placebo	54 weeks	3% intestinal stenosis at wk 54	Hanauer SB et al. [8] (2002)
		192 Pts received IFX (5 mg/kg)		2% intestinal stenosis at wk 54	
		193 Pts received IFX (10 mg/kg)		3% intestinal stenosis at wk 54	
Retrospective	Infliximab	2373 Pts received no IFX		No increased risk of strictures development	Licthenstein GR et al. [9] (2006)
		2785 received IFX		HR:1.114 (95% CI; 0.716–1.734, *p*: 0.63)	
Retrospective	Infliximab	141 Pts with stricturing CD received IFX	40 months (IQR,19–85)	42% (59) needed surgery, *p*: 0.02	Rodriguez-Lago et al. [10] (2020)
Retrospective	Infliximab/Adalimumab	51 CD Pts with colonic or small bowel stenosis	mean: 21.9 months per patient	20 (39.2%) Pts underwent surgery	Alloca M et al. [12] (2017)
				Median time to surgery: 37.9 months	
Retrospective	Infliximab/Adalimumab	84 Pts with stricturing CD	60 months	Surgical or endoscopic approach, hospitalization or treatment discontinuation: 66.6% of Pts within 60 months	Campos C et al. [13] (2017)
Prospective	Adalimumab	97 Pts with symptomatic small bowel stricture received ADA	Median: 3.8 years	64% Steroid-free at wk 24	Bouynik Y et. al. [11] (2018)
				50.7% no surgery after 4 years	
Retrospective	Adalimumab	121 Pts with stricturing CD received ADM	40 months (IQR,19–85)	24% (29/121) needed surgery, *p*: 0.02	Rodriguez-Lago et al. [10] (2020)
Prospective	Ustekinumab	63 Pts with stricturing CD	52 weeks	No predictor factor of steroid-free remission	Hoentjen F et al. [15] (2020)
				(OR: 0.96; 95% CI: 0.4–2.31, *p*: 0.92)	
Retrospective	Vedolizumab	118 Pts with penetrating or stricturing CD	12 months	5 Pts (4.2%) needed surgical approach	Dulai PS et al. [16] (2016)
RCT	Azathioprine Mesalazine	36 Pts with stricturing CD received AZA	36 months	Pts received AZA had fewer hospital admissions (0.70 vs. 1.41; *p* = 0.001), surgical rates (25 vs. 56%; *p* = 0.011) and hospital duration (3.8 vs. 7.7 days; *p* = 0.002)	Chembli JMF et al. [17] (2013)
		36 Pts with stricturing CD received MSZ			

ADA—adalimumab; AZA—azathioprine; CD—Crohn’s disease; IFX—infliximab; MSZ—mesalazine; Pts—patients; HR—hazard ratio; OR—odds ratio; RCT—randomized clinical trial; wk—week; IQR—Interquartile range; CI—confidence interval.

Regarding the role of thiopurines and mesalazine on stricturing CD, a randomized control study suggested that azathioprine administration is associated with better outcomes than mesalazine administration. This study randomized 72 patients with sub-occlusive ileocecal CD who responded to intravenous corticosteroids administrations. They received either mesalamine (3.2 gr/day) or azathioprine (2–3 mg/kg), and the follow-up period was up to 3 years. Patients received AZA had lower re-hospitalizations for surgical approach (25 vs. 56%, respectively; *p* = 0.011), lower number of admissions (0.70 vs. 1.41; *p* = 0.001) and lower mean hospital stay duration (3.8 vs. 7.7 days; *p* = 0.002) [17].

Thus, anti-TNF therapy appears to provide the most significant benefit among available medical treatments; however, a considerable proportion of patients with stricturing CD on the anti-TNF agent will need surgical intervention. In addition, there is limited data about the role of ustekinumab and vedolizumab, and further studies are required.

## 4. Endoscopic Management

In patients with significant stenoses, where medical treatment fails to resolve symptoms, endoscopy and surgery offer effective alternative treatment solutions.

### 4.1. Endoscopic Balloon Dilation

Endoscopic balloon dilation of a strictured segment with a through-the-scope (TTS) balloon is demonstrated to be a minimally invasive, safe, and effective method [18,19] with a small risk of major complications [20,21], enabling the deferral of surgery [22] and thus allowing bowel length conservation.

The technique of endoscopic balloon dilation is rather simple; the stenosis is approached with the endoscope in a retrograde or anterograde manner, the TTS balloon is gently placed inside the stricture and then hydrostatically dilated with water or contrast inside the balloon [23].

Practically, endoscopic balloon dilation can be performed in any part of the gastrointestinal tract, in endoscopically reachable anastomotic or naïve non-malignant strictures [24,25,26]. However, colonic strictures are associated with malignancy in a small but significant percentage of patients [27]. In contrast, the role of endoscopic biopsies in the guidance of therapeutic decisions in patients with colonic strictures is debatable, as tissue samples negative for malignancy were not shown to rule out the presence of dysplasia in 0,8% of CD and 5% of UC patients [27].

Candidates for a successful EBD are patients with partial or resolving symptoms of obstruction, short (≤5 cm), non-angulated, uncomplicated (without extensive ulcerations, fistulization, abscess, perforation) naïve, distal to the duodenum strictures [6,23,28,29]. 

Stricture location, length, and anatomy determine the choice of balloon size, dilation, duration, and dilation approach, namely, graded (a gradual increase of balloon size) or non-graded dilation [30]. Usually, balloon size ranges between 12–20 mm, with smaller sizes used in the small bowel and larger sizes in the colon. However, the use of larger size balloons is associated with perforation and bleeding without any additional benefit [31]. Furthermore, the more distal the stenosis, the larger the diameter of the lumen must be to allow for proper stool passage, which is indeed less of a problem when dilating proximal (small bowel stenosis).

The likelihood of repeated dilations and the requirement for surgery following dilation in a considerable number of patients are both disadvantages of EBD [32]. Moreover, EBD entails a small but significant risk of complications, namely bowel perforation, severe bleeding, sepsis, abscess, fistula formations [33]. In particular, major EBD complications (e.g., perforation, bleeding, or dilation-related surgery) were estimated to occur in 5.3–6.4% of patients undergoing EBD [20,21]. 

A consensus regarding the definition of endoscopic balloon dilation efficacy is lacking. Various studies have estimated efficacy by presenting short-term, as well as long-term results (Table 2). 

A short-term result could be defined as the immediate technical success of the procedure, which the passage of the endoscope can determine through the stricture after endoscopic balloon dilation, marked by symptom resolution. In contrast, long-term efficacy could be defined by the time until a repeated EBD or surgical intervention is required. 

Regarding short-term results, the technical success rate of EBD is estimated at >90%, with clinical efficacy (symptom resolution) varying between 70–80% [20,21]. As it concerns for long-term results, almost 75% of patients with stricturing CD required redilation during a 2-year follow up [25]. Moreover, the cumulative surgery rate during a 5-year follow-up after EBD was 75% [21]. Stricture length (>4 cm) was the most significant factor of stricture recurrence [33,34].

**Table 2 jcm-11-02366-t002:** Studies about endoscopic techniques of stricturing Crohn’s disease.

Study Design	Endoscopic Technique	Number of Patients	Successful Rates (Technical)	Complications	Study (Year of Study)
Meta-analysis	Endoscopic Balloon Dilation	463	94.9%	5.3%	Bettenworth D et al. [26] (2020)
Meta-analysis	Endoscopic Balloon Dilation	1089	90.6%	6.4%	Morar PS et al. [21] (2015)
Meta-analysis	Endoscopic Balloon Dilation	1463	89.1%	2.8%	Bettenworth D et al. [25] (2017)
Retrospective	Endoscopic Stricturotomy	85	100%	3.7%	Lan N et al. [35] (2017)
Retrospective	Endoscopic Stricturotomy	21	100%	8.8%	Lan N et al. [36] (2018)

### 4.2. Endoscopic Stricturotomy

Endoscopic stricturotomy is an innovative technique where an endoscopic needle knife or endoscopic insulated-tip knife is used for the radial incision of a strictured segment. 

Although the available data are scarce, the immediate technical success rate of endoscopic stricturotomy, defined as the successful passage of the endoscope through the strictured segment after stricturotomy, was demonstrated to be high [35] and more efficient than endoscopic balloon dilation [36]. 

Despite the low risk of perforation and low complication rate, endoscopic stricturotomy was shown to have a higher bleeding risk than endoscopic balloon dilation [35,36].

### 4.3. Other Endoscopic Procedures

Although intra-lesional injection with long-acting corticosteroids has emerged as a promising technique for managing stricturing CD, current evidence does not support this practice [28]. In addition, a few small case series have reported benefits with intra-lesional infliximab injections, however the efficacy of this therapeutic strategy requires validation with studies of a higher level of evidence [28].

## 5. Surgical Management

A significant number of CD patients will require surgery to manage fibrostenotic disease, namely resection of the stenotic segment or strictureplasty. A third surgical option, bypass, is mainly indicated for strictures of the gastroduodenal region.

### 5.1. Surgical Resection

Despite the ongoing development of new biologic agents, the risk of intestinal surgery remains high during the disease course. This risk is higher, in patients with a wrong initial diagnosis, delay of diagnosis, and perioperative complication, whereas a number of CD patients will require repeated intestinal resections. Despite the progress in therapeutic options, intestinal-resection associated loss of function bowel length put these patients at risk of short bowel syndrome [37].

Surgical resection is indicated in patients who are not suitable candidates for EBD or stricturotomy, namely when there are complications close to the stricture (fistula, abscess, phlegmon) when there are emergency conditions such as perforation or massive bleeding and when there is a high suspicion of malignancy (e.g., colonic strictures) [38,39].

In addition, early bowel resection could be an option in high-risk patients with isolated ileocecal stricturing CD, as this practice is demonstrated to offer better outcomes when compared to medical treatment regarding the subsequent development of fistula or intestinal obstruction [6].

A laparoscopic approach should be preferred, when possible, as it is associated with reduced morbidity, shorter hospital stay, and reduced iatrogenic complications such as adhesions and hernia formation, whilst it allows improved cosmesis [38]. Past surgery and emergency surgery and malnutrition and anemia, which ideally should be corrected before surgery, were associated with worse outcomes [39]. The main drawback of surgical resection is the reduction of small bowel length, which in case of repeated or extensive surgeries may result in short bowel syndrome. Therefore, the resection should involve the stenotic small bowel segment with margins of 2 cm or less, regardless of the histological activity of the disease in the margins, as it is not demonstrated to affect recurrence [40].

The main types of surgical anastomosis for structuring CD are end-to-end and side-to-side anastomosis. Although the level of evidence is mainly based on a meta-analysis of retrospective studies, with only a small number of randomized control trials, a side-to-side anastomosis was demonstrated to have fewer complications, especially anastomotic leak, and a smaller risk of postoperative recurrence [41,42,43,44,45]. Furthermore, the Kono-S anastomosis and extended mesenteric excision are two surgical methods that have shown promise in lowering postoperative recurrence rates [46,47]. Preoperative treatment with prednisolone >20 mg/day (or other equivalents) increases the risk of anastomotic leakage, surgical site infections and sepsis, whilst the data regarding the use of infliximab-associated postoperative complications are conflicting [39].

### 5.2. Strictureplasty

Strictureplasty is a bowel-length-conserving surgical method, as it allows the widening of the narrowed part of the intestinal stricture without removing an intestinal segment. Moreover, strictureplasty offers the advantage of multiple strictureplasties if necessary, and it can also be combined with surgery. 

Practically every stricture in the jejunum and the ileum outside of endoscopic reach are amenable to strictureplasty [39]. However, strictureplasty is not indicated in patients with long strictures (>68 cm), acute inflammation, local complications (e.g., fistula, abscess, perforation), or suspicion of malignancy. Moreover, strictureplasty is not a therapeutic option in patients with gastric, ileocolonic, or colonic involvement [39].

The main types of strictureplasty are three depending on the length of the intestine:The Heineke–Mikulicz strictureplasty for strictures <10 cm;The Finney strictureplasty for strictures between 10–25 cm;Non-conventional strictureplasties such as the Michelassi strictureplasty for longer strictures up to 68 cm.

In all three types, strictureplasty involves a longitudinal incision along the anti-mesenteric border, and depending on the type, a specific suture method on the strictured area is applied. Despite the fact that it could be difficult to reliably measure a long strictured segment, strictureplasty is not indicated in patients with long strictures (>68 cm).

Strictureplasty complications include small bowel obstruction, sepsis, and other infections, bleeding, progression to cancer of the stenotic segment, mortality, and stricturing reoccurrence, which requires subsequent strictureplasty [48].

Furthermore, non-conventional strictureplasties should be considered in patients with extensive strictures after prior surgical resection in order to avoid complications, such short bowel syndrome [49]. It is worth mentioning that post-operative quality of life of patients after strictureplasty is comparable with resection [50].

## 6. Surgery of Duodenal Strictures

The surgical management of duodenal strictures in Crohn’s disease patients is challenging, as the duodenum is adjacent to the ampulla of Vater, the pancreas, and the mesenteric vasculature; bypass surgery with or without vagotomy, usually with either a gastrojejunostomy or a gastroduodenostomy repair, is one surgical option for controlling gastroduodenal Crohn’s disease [51].

Other options encompass strictureplasty of the duodenum; however, it is not indicated in strictures <10 cm or surgical resection, including pancreas-sparing duodenectomy and pancreaticoduodenectomy, which carry significant complications [52]. Nevertheless, each surgical approach, such as strictureplasty, bypass, and resection, has a distinct purpose, and these alternatives must be customized to the strictures’ characteristics (number, length, and location) and the patient’s comorbidities [53,54].

## 7. Conclusions

Although there have been medical advances in the treatment of CD, the management of fibrostenotic disease remains a significant clinical problem due to the lack of anti-fibrotic agents, affecting the quality of life of CD patients. In recent years, many promising therapies against intestinal fibrotic strictures, such as monoclonal antibodies to IL-36 receptor [55], have been evaluated on animal models; nevertheless, further in vivo trials are needed to assess their efficacy. Currently, TNF inhibitors are demonstrated to be the most effective treatment among the other available agents; however, they are effective only in strictures with a predominantly inflammatory component. Thus, an endoscopic or surgical approach is often necessary and is indicated in patients with fibrotic strictures or non-responding to medical management. Nevertheless, postoperative recurrences are not uncommon and post-operative risk stratification, intense post-operative follow up and the appropriate treatment are mandatory for prevention of postoperative recurrence [56].

## Data Availability

Not applicable.

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
