# Peer review of "Medical, Endoscopic and Surgical Management of Stricturing Crohn’s Disease: Current Clinical Practice"

_jcm, 2022, doi:10.3390/jcm11092366_

Round 1
Reviewer 1 Report
The review aimed to present and analyze the currently available medical, endoscopic, and surgical management of structuring Crohn’s disease. This topic is important for the gastrointestinal system area. I think it will be beneficial for the readers. There is some grammatical mistake. I attached these to the manuscript.

Author Response
Dear editor,
We would like to thank the Reviewers for their useful comments that help our manuscript to reach its full potential.
Hereby, we submit for consideration the revision of our manuscript, titled “Medical, endoscopic and surgical management of stricturing Crohn’s disease: Current clinical practice”. All authors sincerely hope that you will deem it appropriate for publication in your distinguished Journal.
On the revised version of our manuscript a great emphasis has been placed on the instructions and comments placed by the Reviewers and we proceeded with the necessary changes according to the reviewers’ comments.
In particular,
Reviewer1
«There is some grammatical mistake»
All necessary corrections/changes were made and added to the manuscripts.
All revisions in the manuscript have been highlighted. Thank you for receiving our revised manuscript. We appreciate your time and look forward to your response.
On behalf of the co-authors with my great appreciation,
Dimitrios K. Christodoulou
MD, PhD, FEBGH, Professor of Gastroenterology

Reviewer 2 Report
In the paper Medical, endoscopic and surgical management of structuring Crohn’s disease: Current clinical practice (JCM, Fousekis et al 2022), the authors give a structured review of possibilities to medically treat Crohn’s disease associated strictures.
Strengths:
- Concise work of a difficult clinical issue
- Structured literature research
- Clinical practibility
Weaknesses:
- The data per study varies in lengths of follow-up, which is insufficiently documented (tabel 1), and not discussed
- Quality of life isn’t discussed/described
- No qualitative assessment/recommendation is given by the authors, it remains elusive what to do as a clinician
General remarks.
Introduction: CD is a systemic disease, primarily affecting the gastro-intestinal tract, of which the etiology has not been clarified (lines 28-30, page 1).
Page 2, lines 53-58. The role of CRP and fecal calprotectin measurements may be discussed in relation to the techniques described here solely. Line 59, despite is maybe too strong an expression, as what can be expected of anti-inflammatory drugs such as the biologics?
Literature research: I miss an algorithm stating how many articles were identified by the search strategy, by references analysis and how many were screened/excluded.
Medical management line 71-75: Do the authors agree with or support the use of steroids in case of intestinal obstruction? Is the patient tro be characterized before in order to avoid surgery in a patient on high doses of steroids (or biologics??). See remarks on complication risks later in the text as well.
Studies identified (table 1) are being described, but the context and the qualitative judgement of the acquired data is missing. E.g. ref 8 reflects a study that was conducted to assess or deny the risk of generating a stenosis when using infliximab, which was at the time believed to be a risk.
Endoscopic management; do the authors believe that short bowel syndrome is a usual or absolute risk for CD patients nowadays, given the new surgical techniques, the improved imaging techniques and the advent of many new and highly potent drugs? The advocated balloon sizes (line 152-154) may be subject of discussion. The more distal the stenosis the larger the diameter of the lumen must be to allow for proper stool passage, which is indeed less of a problem when dilating proximal (small bowel stenosis). Again, qualitative assessment of literature is lacking.
Surgical management; If the current risks of surgical procedures (including its risks for complications) are diminishing, is earlier application of surgery then warranted (and a better choice than high dose steroid therapy)? Performing stricturoplasty on 68 centimeters of bowel (how the measure that reliably?? And would that be correct whereas 70 cm wouldn’t?) isn’t likely to results in favorable results of a longer period of time (let say several months up till a year). A qualitative comment would be helpful, in my opinion.
General remarks.
In addition, a study or data is no person and therefore is not able to find or to do something (personification). This grammatical error (in Dutch at least) can be found throughout the text.
There are few typos.
Author Response
Dear editor,
We would like to thank the Reviewers for their useful comments that help our manuscript to reach its full potential.
Hereby, we submit for consideration the revision of our manuscript, titled “Medical, endoscopic and surgical management of stricturing Crohn’s disease: Current clinical practice”. All authors sincerely hope that you will deem it appropriate for publication in your distinguished Journal.
On the revised version of our manuscript a great emphasis has been placed on the instructions and comments placed by the Reviewers and we proceeded with the necessary changes according to the reviewers’ comments.
In particular,
Reviewer2
QUERY#1 Weaknesses: A) The data per study varies in lengths of follow-up, which is insufficiently documented (tabel 1), and not discussed. B) Quality of life isn’t discussed/described. C) No qualitative assessment/recommendation is given by the authors, it remains elusive what to do as a clinician
A) We revised the table 1 and we added “Thus, there is increased heterogenity among studies about the role of anti-TNF agents on stricturing CD, because the length of follow-up intervals, inclusion criteria, characteristics of patients and the end-points vary. However, it seems that a portion of patients with stricturing CD seems to respond to anti-TNF therapy, while infliximab administration appears to be not associated with strictures development.”
The answer can be found on page 6 line 132-137
B) “Quality of life isn’t discussed/described”. We added comments about the quality of life of patients with Crohn’s disease and how surgical treatment affect it “It is worth mentioned that post-operative quality of life of patients after strictureplasty is comparable with resection.” To our knowledge, the literature about the affect of endoscopic and surgical approach on quality of patients with stenotic Crohn’s disease is limited.
The answer can be found on page 13 line 299-300
C) “No qualitative assessment/recommendation is given by the authors, it remains elusive what to do as a clinician”. In the conclusion, we comment “TNF inhibitors are demonstrated to be the most effective treatment among the other available agents; however, they are effective only in strictures with a predominantly inflammatory component. Thus, an endoscopic or surgical approach is often necessary and is indicated in patients with fibrotic strictures or non-responding to medical management. Nevertheless, postoperative recurrences are not uncommon and post-operative risk stratification, intense post-operative follow-up and the appropriate treatment are mandatory for prevention of postoperative recurrence”. In the text, we present and the table 2 we present the indications each endoscopic and surgical technique.
The answer can be found on page 14 line 321-328
QUERY#2 «Introduction: CD is a systemic disease, primarily affecting the gastro-intestinal tract, of which the etiology has not been clarified (lines 28-30, page 1).»
The necessary changes where added as indicated by the reviewer and can be found on page 3 line 46-47.
QUERY#3 «Page 2, lines 53-58. The role of CRP and fecal calprotectin measurements may be discussed in relation to the techniques described here solely. Line 59, despite is maybe too strong an expression, as what can be expected of anti-inflammatory drugs such as the biologics?»
In response to your query an addition was made to the original manuscript which can be found on page 4 line 76-83:
”On the other hand, inflammatory markers, such as CRP and fecal calprotectin, may reflect CD activity, contributing to distinguish between fibrotic and inflammatory strictures without to be strictly specific (5). Thus, clinicians should take account the clinical course of CD, inflammatory biomarkers and imaging techniques in order to diagnose a fibrotic or inflammatory stenosis. The ongoing development of new biological agents offers more therapeutic options with different action mechanisms, however; the management of stricturing CD remains a challenging task.”
QUERY#4 «Literature research: I miss an algorithm stating how many articles were identified by the search strategy, by references analysis and how many were screened/excluded.»
This manuscript is a narrative review; hence, we did not document the articles selection process and there is not a flow diagram. In the section of literature research, we added: “Only articles in the English language were used. We extensively examined the abstracts of manuscripts and identified the most relevant articles.” which can be found on page 5 line 92-94
QUERY#5 Medical management line 71-75: Do the authors agree with or support the use of steroids in case of intestinal obstruction? Is the patient tro be characterized before in order to avoid surgery in a patient on high doses of steroids (or biologics??). See remarks on complication risks later in the text as well.
In response to your query we made an addition to the manuscript text which can be found on page 5 line 100-103: “Corticosteroids should be administered in patients without signs of peritonitis and suspicious of perforation. A surgical approach is required if conservative management is failed and/or there are features of abscess, phlegmon and malignancy.”
QUERY#6 Studies identified (table 1) are being described, but the context and the qualitative judgement of the acquired data is missing. E.g. ref 8 reflects a study that was conducted to assess or deny the risk of generating a stenosis when using infliximab, which was at the time believed to be a risk
In response to your query we made an addition to the manuscript text which can be found on page 6 lines 132-137: “Thus, there is increased heterogenity among studies about the role of anti-TNF agents on stricturing CD, because the length of follow-up intervals, inclusion criteria, characteristics of patients and the end-points vary. However, it seems that a portion of patients with stricturing CD seems to respond to anti-TNF therapy, while infliximab administration appears to be not associated with strictures development.”
QUERY#7 Endoscopic management; do the authors believe that short bowel syndrome is a usual or absolute risk for CD patients nowadays, given the new surgical techniques, the improved imaging techniques and the advent of many new and highly potent drugs?
In response to your query we made an addition to the manuscript text which can be found on page 11 line 237-242: “Despite the ongoing development of new biologic agents, the risk of intestinal surgery remains high during the disease course. This risk is higher, in patients with a wrong initial diagnosis, delay of diagnosis, and perioperative complication, whereas, a number of CD patients will require repeated intestinal resections. Despite the progress in therapeutic options, intestinal-resection associated loss of function bowel length put these patients at risk of short bowel syndrome”.
The advocated balloon sizes (line 152-154) may be subject of discussion. The more distal the stenosis the larger the diameter of the lumen must be to allow for proper stool passage, which is indeed less of a problem when dilating proximal (small bowel stenosis). Again, qualitative assessment of literature is lacking.
In response to your query we made an addition to the manuscript text which can be found on page 9 line 192-194: “Furthermore, the more distal the stenosis the larger the diameter of the lumen must be to allow for proper stool passage, which is indeed less of a problem when dilating proximal (small bowel stenosis).”.
QUERY#8 Surgical management; If the current risks of surgical procedures (including its risks for complications) are diminishing, is earlier application of surgery then warranted (and a better choice than high dose steroid therapy)?
In response to your query we made an addition to the manuscript text which can be found on page 5 line 103-106: “In addition, early bowel resection could be an option in high-risk patients with isolated ileocecal stricturing CD, as this practice is demonstrated to offer better outcomes when compared to medical treatment regarding the subsequent development of fistula or intestinal obstruction”.
Performing stricturoplasty on 68 centimeters of bowel (how the measure that reliably?? And would that be correct whereas 70 cm wouldn’t?) isn’t likely to results in favorable results of a longer period of time (let say several months up till a year). A qualitative comment would be helpful, in my opinion.
In response to your query we made an addition to the manuscript text which can be found on page 13 line 291-293: “Despite the fact that it could be difficult to reliably measure a long strictured segmen, stricturoplasty is not indicated in patients with long strictures (>68cm)”.
QUERY #9 General remarks.
In addition, a study or data is no person and therefore is not able to find or to do something (personification). This grammatical error (in Dutch at least) can be found throughout the text. There are few typos.
All necessary corrections/changes were made and added to the manuscripts.
All revisions in the manuscript have been highlighted. Thank you for receiving our revised manuscript. We appreciate your time and look forward to your response.
On behalf of the co-authors with my great appreciation,
Dimitrios K. Christodoulou
MD, PhD, FEBGH, Professor of Gastroenterology
